# Establishing Occupational Therapy Needs: A Semi-Structured Interview with Hereditary Transthyretin Amyloidosis Patients

**DOI:** 10.3390/ijerph191811721

**Published:** 2022-09-17

**Authors:** Aina Gayà-Barroso, Juan González-Moreno, Adrián Rodríguez, Tomás Ripoll-Vera, Inés Losada-López, Margarita Gili, Milena Paneque, Eugenia Cisneros-Barroso

**Affiliations:** 1Internal Medicine Department, Son Llàtzer University Hospital, 07198 Palma de Mallorca, Spain; 2Health Research Institute of the Balearic Islands (IdISBa), Son Llàtzer University Hospital, 07198 Palma de Mallorca, Spain; 3Cardiology Department, Son Llàtzer University Hospital, 07198 Palma de Mallorca, Spain; 4Department of Psychology, University of Balearic Islands, Health Research Institute of the Balearic Islands (IdISBa), 07120 Palma de Mallorca, Spain; 5Institute for Research and Innovation in Health (i3S), University of Porto, 4099-002 Porto, Portugal; 6Center for Predictive and Preventive Genetics, Institute for Molecular and Cell Biology (CGPP-IBMC), 4200-135 Porto, Portugal; 7Institute of Biomedical Sciences Abel Salazar, University of Porto, 4099-002 Porto, Portugal

**Keywords:** transthyretin amyloidosis, occupational therapy, polyneuropathy, semi-structured interview, intervention

## Abstract

The purpose of this study was to explore the occupational performance and needs of patients with hereditary transthyretin amyloidosis (ATTRv). A semi-structured interview was conducted by an occupational therapist with 44 patients with Val50Met-ATTRv recruited through patient associations. The interview addressed three related dimensions. The first one, the physical dimension, was evaluated using the Spanish versions of the Barthel Index, the Lawton and Brody scale, and the Norfolk questionnaire; the second one, the psychological dimension, was assessed with the Warwick–Edinburgh Mental Well-Being Scale and the SF-36 questionnaire; and the third dimension, the occupational performance, was assessed through unstructured questions on daily occupations, work, roles, and hobbies given the lack of standardized scales. Twenty participants (45.4%) responded that the disease had affected their basic activities of daily living, twenty- four (54.5%) perceived an impact on their instrumental activities of daily living, and all the participants reported that the disease symptoms had affected their ability to perform advanced activities as well as their employment status. Only three patients (6.8%) reported a lack of psychological impairment following disease diagnosis. These findings suggest that a semi-structured interview conducted by an occupational therapist can provide essential information that should be considered for the implementation of occupational therapy programs targeting patients living with a diagnosis of ATTRv.

## 1. Introduction

Hereditary transthyretin amyloidosis (ATTRv) is defined as a heterogeneous disease. The main characteristic is amyloid deposition, which is derived from the accumulation of unstable conformations of the transthyretin (TTR) protein [1,2,3,4,5,6,7,8]. There are more than 140 mutations in the *TTR* gene already described. The most common variant is known as V50M and is particularly prevalent in the two largest foci of the disease in Spain, which are found in Majorca (Balearic Islands) and Valverde del Camino (Huelva), where ATTRv is described as an endemic disease [9,10]. The symptomatology of this rare disease is variable among patients from different geographical areas, but the most severe presentations are debilitating and life-threatening and are associated with physical, psychological, occupational, and social symptoms. ATTRv is usually characterized by neuropathic symptoms mainly related to peripheral damage—at both sensory and motor levels—and autonomic neuropathy [11,12]. Gastrointestinal impairment, cardiomyopathy, ocular symptoms resulting from amyloid deposition in the eye, and nephropathy frequently appear in association with neurological impairment [8]. Symptoms are progressive and disabling throughout the four main polyneuropathy stages of the disease (I, II, III, IV), with severe impairment in the IV stage [10,11]. The first phase is characterized by mild autonomic dysfunction and sensory impairment in the lower limbs. In the second phase, as a progressive disease, with autonomic dysfunction and sensory-motor impairment of the upper limbs, and in the third phase, dysfunction is severe, and even complete paralysis may develop in the fourth phase [3,4,5,6,7,8,9,10,11].

The limitations in daily activities faced by ATTRv patients in all stages of the disease—although to a greater extent in advanced stages (III, IV)—have been described as difficulties impairing personal independence in self-care skills and self-determination [12]. An individual’s occupational performance is essential for a balanced daily life [12,13,14,15,16]; by developing specific interventions for these patients, occupational therapy (OT) can play an important role in reducing activity limitations, improving personal autonomy, empowering patients to overcome disease-related barriers and constitutes a pioneer project, due to the fact that up to now no psychosocial interventions have been carried out on ATTRv patients [11]. OT can intervene using specific techniques and programs that help develop, maintain, or improve the performance of daily living activities in patients with a diagnosis of the disease by compensating for dysfunctions and promoting occupational health and general well-being [13,14,15,16,17].

The OT practice framework classifies individual functions in occupational areas (physical, psychological, and occupational) through semi-structured interviews. In the present study, the Barthel Index (BI), Lawton and Brody Instrumental Activities of Daily Living (IADL) scale were used to obtain information about patients’ physical function, Warwick–Edinburgh Mental Well-Being Scale (WEMWBS), SF-36 questionnaire were used to obtain relevant information about patients’ mental health and psychological function, and Norfolk Quality of Life-Diabetic Neuropathy (Norfolk QoL-DN) questionnaire—already validated for ATTRv patients [18]—were used to review the most common symptomatology and physical function. In addition, given the lack of standardized scales focusing on occupational issues, unstructured questions, as a part of the semi-structured interview, were done. Occupational topics for collecting information pre- and post-diagnosis were used by an occupational therapist. As the main objective of an occupational intervention is to improve patient independence in performing daily activities by reducing limitations and providing training to achieve autonomy [12,13,14,15,16,17], this study aimed to conduct a semi-structured interview in patients with ATTRv developed to understand their occupational performance and needs better.

This study is part of a set of three studies. Accessibility of patients to OT services was assessed in the first one; the second and present study aimed to investigate how to determine occupational needs; and finally, a third and future study will seek to investigate the impact and relevance of occupational interventions in ATTRv patients according to the symptoms. The aim of these three studies is to demonstrate the positive impact that OT may have on individuals with ATTRv, as shown in other rare diseases, such as Charcot–Marie–Tooth Disease, for which there is clear evidence that OT plays an important role [16].

## 2. Methods

### 2.1. Study Design and Approval

A descriptive qualitative, cross-sectional, and non-interventional online study was conducted. An experienced occupational therapist designed a semi-structured interview and conducted all the interviews. Participants were allowed to have assistance in answering questions when needed because of the severity of their symptoms. The semi-structured interview was the chosen research method given its low cost and the simplicity of the data analysis required.

### 2.2. Study Population

Patients aged 18 years or older and diagnosed with ATTRv were eligible for the study. Participants were recruited through patient association websites and from the Son Llàtzer University Hospital database. Patients who were interested in the study and fulfilled the inclusion criteria contacted the occupational therapist. Caregivers or asymptomatic carriers of pathogenic mutations in the *TTR* gene were excluded from the study.

### 2.3. Data Collection

The semi-structured interview, designed by an Occupational Therapist, was reviewed by the medical research team from Son Llàtzer University Hospital, formed by doctors and the Clinical Research Coordinators and ABEA, whose review was carried out by the board members. Many questions allowed for open-ended answers, particularly in the occupational dimension, allowing participants to provide in-depth answers. The rationale for using this semi-structured interview was to obtain an insight into whether patients with ATTRv are limited in the performance of their daily living activities.

A brief pilot study was carried out with the board members of ABEA to make modifications before starting the study with the rest of the sample. The patients included in the pilot study underwent a semi-structured interview and were asked for feedback at the end. 

The study was conducted from 20 February 2021 to 28 February 2022. A demographic questionnaire, five standardized scales, and a five-item questionnaire on occupational issues were completed by the patients during an online interview conducted using Webex or Zoom. This approach was selected to generate a pleasant atmosphere in which patients would be more likely to provide detailed information, and also because of the COVID-19 pandemic restrictions. Each interview lasted from 40 to 50 min and concluded when the patient was satisfied with all the answers. Because of the online nature of the study, patients provided their oral consent to participate in the study by confirming that they had understood the patient information provided and that they freely agreed to participate in this study.

### 2.4. Instrument

The semi-structured interview was organized in two parts. The first part was a questionnaire consisting of 12 demographic items, including general data like age, gender, age at diagnosis, marital status, offspring, liver transplant, pharmacological treatment, and symptoms, and a set of questions related to occupation, in which the patients were asked about their occupational situation before and after the diagnosis. The second part comprised the Spanish versions of the standardized scales described below.

For the physical activity, Barthel Index BI, which assesses ten items (feeding, bathing, grooming, dressing, bowels, bladder, toilet use, transfers, mobility, and stairs) and has a total score of 100, was used first. A total score between 0 and 20 suggests total dependence for the performance of basic activities of daily living, a score of 21 to 60 shows severe dependence, of 61 to 90, moderate dependence, of 91 to 99, mild dependence, and of 100, independence [18]. The Lawton and Brody IADL scale was then used to assess the participants’ functional capacity through the following eight items: the ability to use the telephone, make purchases, prepare food, take care of the house, wash clothes, and use means of transport, and responsibility regarding medication and administration of their economy. Each item is assigned a numeric value of 1 (independent) or 0 (dependent). The final score is the sum of the values of all the responses and ranges from 0 (maximum dependence) to 8 (total independence) [19]. Finally, the Norfolk QoL-DN questionnaire consists of five domains (35 scored items) organized thematically into five parts assessed physical functioning/large fiber neuropathy, activities of daily life, symptoms, small fiber neuropathy, and autonomic neuropathy [20].

To evaluate mental health, the WEMWBS was used, capturing a wide range of well-being-related questions, including affective-emotional aspects, cognitive-evaluative dimensions, and psychological functioning, in a form that is short enough to be used in population-level surveys. The scale consists of 14 items that are evaluated based on a 5-point Likert scale yielding a minimum score of 14 and a maximum score of 70 [21]. The SF-36 questionnaire evaluated general well-being, including nine items on well-being perception: physical functioning, role limitations due to physical problems, role limitations due to emotional problems, fatigue, emotional well-being, social functioning, pain, general well-being, and changes in health [22].

Lastly, the occupational dimension was assessed based on unstructured questions, which are part of the script of the semi-structured interview. Questions related to work, roles, and hobbies before and after diagnosis were done to compare and obtain relevant data for this study. The approach based on this combined use of standardized scales and an occupational interview was followed as it provided essential information on patients’ occupational status. The questions asked to assess the occupational status are summarized in Table 1 and the framework used for planning the semi-structured interview is in Table 2. All the information obtained from each field of study was used to make a global profile of each patient.

### 2.5. Statistical Analysis

A descriptive analysis was performed to explore the impact of the disease on the physical, psychological, and occupational variables included in the study. Categorical variables were expressed using numbers and percentages. Mean and standard deviation for continuous variables after assessing for normality. The Statistical Package for the Social Sciences (v.23), version 23, has been used specifically for this study.

### 2.6. Ethical Considerations

Ethical approval was granted by the Ethics Committee of the Balearic Islands and the Research Commission of Hospital Universitario Son Llàtzer (Decision number: IB 4587/21 PI). Participants in the semi-structured interview were informed that involvement in the study was confidential, anonymous, and voluntary. Informed consent was collected verbally due to the virtual nature of the interviews to agree with the social distancing measures applied as a cause of the epidemiologic situation.

## 3. Results

Forty-four patients were recruited between February 2021 and February 2022; patients included in the study were those who were contacted with ABEA or with the research group of Hospital Son Llàtzer Patients were predominantly women (30; 68.18%) with a mean age of 53.06 (±SD = 13.9) years. The mean age at disease onset was 43.51 (±SD = 16.3) years. Most patients lived with a partner (27; 61.36%), while 9 (20.45%) were single.

The assessment of occupational aspects revealed that most patients were not working because of their ATTRv symptoms 16 (36.4%) or were retired 13 (29.5%), with only 15 (34.01%) who were employed. The OT semi-structured interview contained two questions regarding leisure time. All participants had had at least one hobby, such as walking, pottery, reading, or cooking, before the diagnosis. After the diagnosis, 3 (6.8%) explained that they had abandoned their hobbies and leisure activities and they did not occupy their leisure time at all because of the lack of motivation and the impact of their symptoms, and 13 (29.5%) had reduced their leisure activities because of the symptoms, the COVID-19 pandemic, and the lack of motivation. A further 26 patients (59.1%) explained that they spent their time as they did before the diagnosis, and only 2 (4.5%) stated that their leisure time had improved as the diagnosis as they were involved in more leisure activities despite their symptoms.

Physical activity was evaluated using the BI and Lawton and Brody IADL scale (Figure 1). 38 (86.36%) subjects were independent for the performance of basic activities of daily living (BADL) and were in the early stages of the disease (I, II), whereas 3 (7%) were slightly dependent, and 3 (7%) were moderately dependent, the 6 of them in advanced stages of the disease (III, IV). 22 (50%) patients did not have a maximum score in the Lawton and Brody test, meaning that they had difficulties in performing their IADL because of ATTRv symptoms.

The results obtained with the WEMWBS scale are shown in Figure 2. Participants were asked about the impact of the diagnosis on their mental well-being, and 36 (82%) declared that the diagnosis had altered in some way their mental health.

In the Norfolk QoL-DN questionnaire, all patients (100%) blamed their difficulties in daily life activities on their ATTRv symptoms even though they continued performing them to a large extent (Figure 3). The most commonly reported symptoms were pain and loss of sensation in the feet and legs, difficulty with fine motor skills, and tiredness. Four (8%) patients surveyed obtained a total score between 80 and 100, 5 patients (11%) between 79 and 60, and 17 (39%)—the largest group—obtained a score between 40 and 59.

SF-36 questionnaire scores are shown in Figure 4. General well-being, social function, and emotional well-being were the three most affected items. During the interview, the patients stated that the diagnosis had had a huge impact on their daily life and that sometimes this was difficult to measure using grading scales and numerical scores. Specifically, in relation to the SF-36 questionnaire, the patients reported difficulties scoring emotions and objective situations. The patients surveyed declared that the diagnosis had clearly had a negative impact on their lives but that it was difficult to measure all the changes experienced with numbers. In addition, patients reported knowing that their current situation was liable to change and that it would worsen as the disease progressed. Of the nine areas assessed with this questionnaire, all the patients (44; 100%) reported impairment in three areas, 43 (98%) reported some changes in four areas, and the two areas with the lowest scores were physical limitations and emotional limitations (20: 45% and 25; 57%, respectively).

## 4. Discussion

Our findings raise important questions that must be addressed.

Firstly, although high scores were obtained for BADL and IADL, these are likely to reflect the fact that most patients were in stage I of the disease. As the disease is heterogeneous and progressive [1,2,3,4,5,6,7], this situation of independence is expected to change with time. Indeed, a related study found that more than 80% of patients reported perceiving the diagnosis or identification of the disease as a life-changing event that altered their plans and daily occupations [10]. The patients interviewed in our study expressed their concern regarding daily life autonomy and their awareness about progressively losing their occupational skills and independence.

Secondly, although standardized scales were used in the present study, the lack of knowledge and specific scales and the scarcity of literature on occupational performance in patients with ATTRv made it difficult to collect comprehensive data on basic occupational conditions. Nevertheless, the SF-36 questionnaire and WEMWBS scale scores showed that ATTRv diagnosis has a significant impact on certain aspects of general well-being, such as emotional well-being, social functioning, or general well-being, revealing the crucial impact of ATTRv on patients’ quality of life. Accordingly, diagnosis of ATTRv is likely to represent a challenging situation for occupational therapists whose profession is focused on the promotion of health and well-being through occupational interventions [13]. In line with our findings, other studies that have focused on the psychosocial impact of ATTRv suggest that the most significant impact is essentially on the quality of life and patients’ mental health once the symptoms become noticeable [11]. However, there is no relevant information available on the role of the occupational therapist in ATTRv and the usefulness of conducting a semi-structured interview [11]. In our study, the BI and the Lowton and Brody IADL scale scores indicated that 38 (86.36%) and 22 (50%) participants, respectively, reported being independent. This difference in the results is due to the difficulty in assessing the activities examined with each instrument. The activities evaluated with the BI are more basic and simpler than the ones assessed with the Lawton and Brody scale. Among patients who reported an impact of the disease on their work or study plans, 16 (36.36%) indicated that they had had to stop working due to pain, stress, or disability. Furthermore, leisure activities and daily occupations had been altered since diagnosis in a high percentage of patients 13 (29.5%). This represents essential information that should be taken into account for the development of OT programs. The first OT interviews and interventions should take place, even if the patients are in the early stages (I, II) of ATTRv, to maintain their independence as long as possible [23,24,25]. Early effective interventions are necessary to maximize the benefits of OT in these patients [16]. Similar studies on OT and rare diseases have shown that patients who have been included in OT intervention programs can experience significant benefits in their daily lives. OT in diseases like CMT highlights the importance of OT interventions to treat the levels of dependency. Therefore, this information could be extrapolated to ATTRv disease [16]. Our findings demonstrate the significant impact of ATTRv diagnosis on occupational performance as well as the need to develop appropriate OT educational programs aimed at multidisciplinary teams of healthcare professionals, occupational therapists, and ATTRv patients to ensure access to OT services and personalized interventions that may satisfy these patients’ specific occupational needs in a timely manner [23,24,25,26]. This result is consistent with the literature [23,24,25,26] after making a personal point of view.

## 5. Limitations

The small study sample size, which only allowed for an insight into a particular group of patients with ATTRv—mainly in stages I and II of the disease—may be due to the recruitment method used: patients were recruited through the Balearic ATTRv patient association and Hospital Son Llàtzer, and therefore involved only a small proportion of the total Spanish ATTRv patient population. A larger sample size may have revealed different aspects or provided additional relevant information. However, we believe the findings obtained through the analysis of the data collected could be representative.

Additionally, an obvious limitation was the lack of standardized scales to measure variables such as occupational performance properly. Previous research in OT and ATTRv shows a significant scarcity of available literature on these two subjects. Nonetheless, the measuring tools used in this study appear to have provided relevant findings.

## 6. Conclusions

In conclusion, our findings reveal that ATTRv is debilitating not only physically but also psychologically and occupationally in the studied sample. Most patients included in this study were in stage I, therefore showing a high level of independence, but because of the progressive nature of the disease, these patients report being aware of their limiting situation and should receive OT interventions to delay dependency. The data obtained through the semi-structured interviews performed may provide essential information for future specific OT programs, as well as represent a bibliographical contribution in terms of ATTRV and Occupational Therapy.

## Figures and Tables

**Figure 1 ijerph-19-11721-f001:**
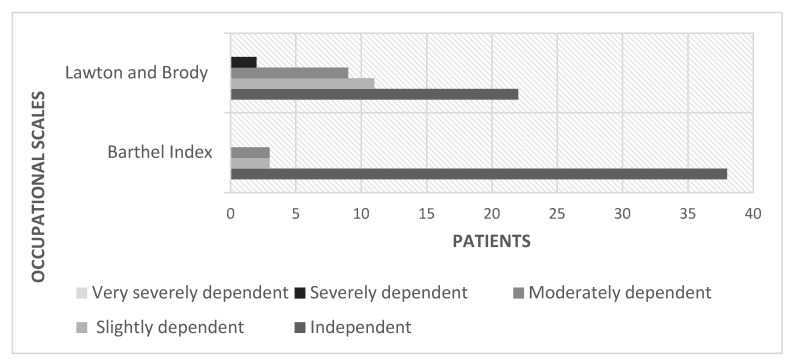
The proportion of patients with difficulties in basic and instrumental activities of daily living.

**Figure 2 ijerph-19-11721-f002:**
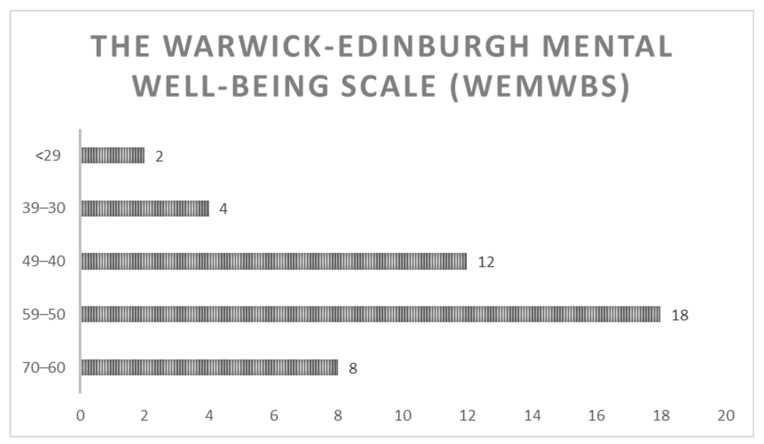
Warwick–Edinburgh Mental Well-Being Scale (WEMWBS) results. The results show the number of patients who have obtained scores between the explained ranges; higher scores represent better results.

**Figure 3 ijerph-19-11721-f003:**
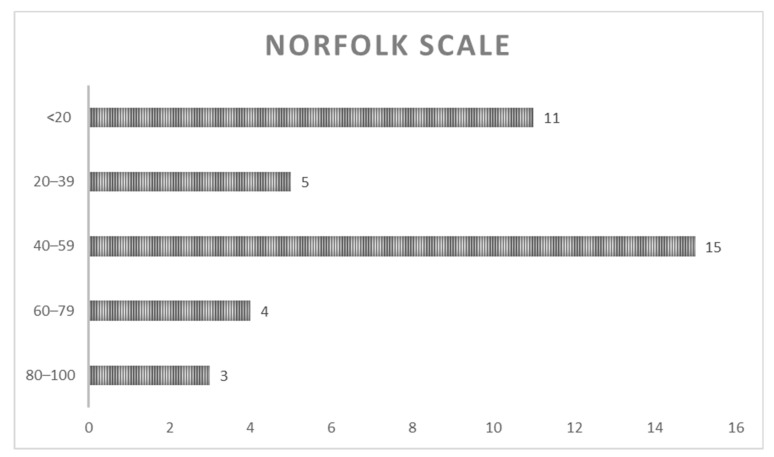
Norfolk QoL-DN questionnaire scores. This figure represents the number of patients who have obtained scores within each range; lower scores represent better results.

**Figure 4 ijerph-19-11721-f004:**
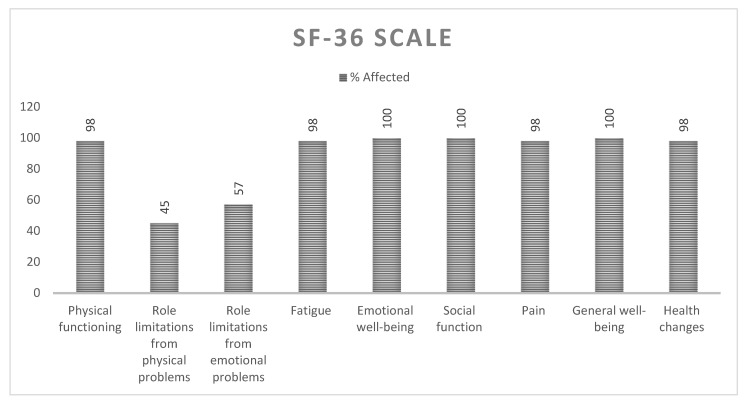
SF-36 questionnaire results.

**Table 1 ijerph-19-11721-t001:** Occupational interview questions.

How would you define your work situation before the diagnosis of ATTRv?
How would you define your work situation after receiving the diagnosis?
Could you list the most significant hobbies before diagnosis? And after? (Briefly explain what they consist of)
Do you consider that you are less busy since the diagnosis (routines)? Why?
Bearing in mind that roles are behaviors expected by society, shaped by culture, and conceptualized and defined by an individual, do you think the diagnosis of ATTRv has affected your roles in daily life?

**Table 2 ijerph-19-11721-t002:** Semi-structured interview framework.

First section:
- Demographic data - Occupational questions
Second section:
- Barthel Index and The Lawton and Brody scale: physical function - The Warwick–Edinburgh mental well-being scale and SF 36: psychological function - Norfolk QoL-DN questionnaire: physical function and main symptoms

## Data Availability

The dataset generated during and/or analyzed during the current study is available from the corresponding author upon reasonable request.

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
