# Peer review of "Establishing Occupational Therapy Needs: A Semi-Structured Interview with Hereditary Transthyretin Amyloidosis Patients"

_ijerph, 2022, doi:10.3390/ijerph191811721_

Round 1

Reviewer 1 Report

No written informed consent has been obtained by the patients. Otherwise, the paper contains some interesting points of interest, especially about the use of a questionnaire about the occupational aspects of ATTR-v. However, even the questionnaire is worth to be improved, in my opinion, including, i.e., the reasons why the activities are reduced: cardiological symptoms, strenght and balance impairment or dysautonomia are some of the most common limitations for patients and the reader could find interesting to know which among them mostly affects activities.

Finally, a better literature review is recommended, including some milestone articles about the disease. 

Reviewer 2 Report

Dear authors,
It is a meaningful topic to discuss the needs and obstacles of special diseases in the employment process. However, the main purpose of the results of the article is to provide readers with the research contribution and knowledge gained after reading this manuscript. Therefore, it is recommended that the author revise the manuscript according to the suggestions to improve the visualization of the manuscript.
1. About the layout of the manuscript format
This manuscript has too many chapters and is suggested to be shortened.
Normally, no more than 6 chapters.
2. About Introduction and Relevance
a. Although the author has certain descriptions of the disease and related treatment and employment issues. And seems to clearly point out the importance of this research" This section is in "2. Relevance".
Because the research direction has been clearly stated at the end of the Introduction. Yet again at Relevance there is talk of the importance of research.
I suggest re-discussion after summarizing the two parts.
b. Regarding Relevance, the author seems to want to summarize the research topic. However, it has not been able to explain why the three scales such as "WEMWBS", "Norfolk QoL-DN", and "SF-36" are used.
I suggest a supplementary statement on the appropriateness of the scale to the research subjects.
Furthermore, the use of data, literature or questionnaires is subject to verification after application or adaptation. Because the scales used by predecessors (research topics or objects) may not be applicable to later generations.
Therefore, I hope the author can supplement the results of the test of trustworthiness and filial piety of this scale.
3. About Methods
a. This study appears to be a mixed study. Because the author adopts semi-structured interviews, three scales are used for questionnaire sampling. So how did the author integrate, integrate and analyze the two data? Please provide additional explanation.
b. Although the researcher conducts sampling and surveys in a variety of research methods, this increases the rigor of the research. But I don't think it's easy for readers to understand the research process.
I therefore recommend that the manuscript be supplemented with a research framework or flowchart.
4. About Results
a. The author presents the results using bar and pie charts. I think this is a good way. However, the colors in the pictures are not easy to read and distinguish.
I recommend color? (if the journal allows it) or other appearance to identify.
b. In addition, the font is too small, and the numbers are not easy to read.
c. Regarding Figure 4, 45% of the description has been truncated by the picture. Please fix it.
5. About Discussion
The discussion should present the authors' views on the results of the analysis. If the author needs to verify, it is recommended to add "This result is consistent with the literature [14] after making a personal point of view. Please correct the writing of this chapter.
6. About Limitations
It is recommended to move the restriction to ''Methods'. Or merge in the discussion of "Conclusion". And supplementary recommendations for future research due to limitations.
7. About Conclusion
This paragraph is very concise, not bad. It would be better to make recommendations based on the limitations of the discussion and research itself.
good luck,

Reviewer 3 Report

Dear Authors,

Manuscript shows the occupational performance and needs of patients with hereditary transthyretin amyloidosis. I have some considerations for you.

-          Abstract

Line 27: “24” you have written before and after the numbers with words, follow the same format.

-          Introduction

You have explained the disease but only the OT treatment, is there any evidence about other treatments for this disease that could influence the evolution of the ATTRv?

Which is its prevalence? More information about the disease should be added.

-          Relevance

Line 83: “multidisciplinary team? Which professionals take part of this team?

I think that introduction and relevance should be in one section. Rewrite the aims to avoid duplicities.

-          Methods

Ethical considerations section should be added at the end of the methods. Also, the ethical approval could be removed to this section. Also the verbal consent and the information about the procedure of the study should be added to this section.

Who create the scale? Only doctors? Who made up the medical team? Was there any expert on the subject or any external opinion from outside the hospital?

It is a new semi-structured scale, who decide the questions (excluding medical team) any occupational therapist, any patient? Dis you implement a pilot study to confirm the correct understanding of the questions by the patients or potential bias that should be corrected at the beginning of the implementation?

I am not a statistic professional but more deep analysis could be done than descriptive analysis.

Line 184 “occupied” must be change for spent. English editing should be followed.

Line 188 “Thirty-eight” number or words but follow the same format.

Line 204, 205: same comment about formats.

-          Results

A sample size calculation should be added to justify the number of participants.

-          Discussion

Although the literature for this disease is scarce you have mention another rare disease, you should compare your results with other disease to highlight your OT treatment. On the other side, you never mention in which the OT treatment consist, you should add a brief explanation about this profession (more than the sentence at the introduction).

-          Author contributions, conflict of interest, and data availability section should be added before references.

-          References:

Follow mdpi rules to adapt the format of the reference for this journal.

In my opinion, the strength of the study is its novelty but methodological limitations have to be taken into account. Objectives of the study should be rewritten, the manuscript shows the functional limitations and symptoms prevalence they experience through a tele-evaluation online, but it could be said that this scale could be used in future research after this implementation as you mention in your limitations. The first step to do that is the scale creation by an expert consensus, followed by a pilot study and finally the validation procedure checking the psychometric properties of the scale you proposed. In this sense title, abstract and discussion should be adapted. If you want to demonstrated the usefulness of OT in the treatment of these patients, you should implement the treatment and after that the reevaluation to evaluate the change that this treatment has achieved like you said in your future study.

Round 2

Reviewer 2 Report

Dear author

Glad to see a revised manuscript submitted. This manuscript is more intuitive and clearer than the previous version. I believe that this manuscript is presented in a clear, readable manner that provides readers with an easy-to-understand understanding of the significance and contributions of this research.
But there are still the following shortcomings, which need to be adjusted. E.g:
1. After the author cancels the literature discussion in Chapter 2, the order of the chapters is wrong.
2. Figures 2 and 3 are not clear.

I think the authors will improve the visualization of the manuscript when they refine these adjustments.
good luck,

Author Response

First of all, we would like to thank you to the reviewers and editor for taking the time of reviewing our manuscript. We are now submitting a revised version of our manuscript “Establishing occupational therapy needs: A Semi-Structured Interview with Hereditary Transthyretin Amyloidosis patients”.

We have revised the manuscript with respect to specific issues raised by the reviewers as well as attempting to clarify points that were in the original manuscript but not appreciated by the readers. 

We really hope that the current version is acceptable for publication. 

Our responses to the referees’ specific comments are noted below.

Dear author

Glad to see a revised manuscript submitted. This manuscript is more intuitive and clearer than the previous version. I believe that this manuscript is presented in a clear, readable manner that provides readers with an easy-to-understand understanding of the significance and contributions of this research.

Thanks for the suggestions. We have modified all the mentioned sections.

But there are still the following shortcomings, which need to be adjusted. E.g: 
1. After the author cancels the literature discussion in Chapter 2, the order of the chapters is wrong.

Answer:

According to the review comment, the order has been modified.

  1. Figures 2 and 3 are not clear.

Answer:

Following the recommendations of the review figures 2 and 3 have been restructured.

Thank you very much again for taking your time in reviewing our manuscript and for the valous comments that have allowed us to significantilly improve it.

All the best

Eugenia

Reviewer 3 Report

Dear authors, 

Thank you to consider my suggestions. I have a better understanding of your work.

Good job and good luck with its continuation.

Author Response

First of all, we would like to thank you to the reviewers and editor for taking the time of reviewing our manuscript. We are now submitting a revised version of our manuscript “Establishing occupational therapy needs: A Semi-Structured Interview with Hereditary Transthyretin Amyloidosis patients”.

We have revised the manuscript with respect to specific issues raised by the reviewers as well as attempting to clarify points that were in the original manuscript but not appreciated by the readers. 

We really hope that the current version is acceptable for publication. 

Our responses to the referees’ specific comments are noted below.

Comments and Suggestions for Authors:

Dear authors, 

Thank you to consider my suggestions. I have a better understanding of your work.

Good job and good luck with its continuation.

Answer:

Thank you very much for all the suggestions made.
